# Rebenchmarking Unsupervised Monocular 3D Occupancy Prediction

## Abstract

Inferring the 3D structure from a single image, particularly in occluded regions, remains a fundamental yet unsolved challenge in vision-centric autonomous driving. Existing unsupervised approaches typically train a neural radiance field and treat the network outputs as occupancy probabilities during evaluation, overlooking the inconsistency between training and evaluation protocols. Moreover, the prevalent use of 2D ground truth fails to reveal the inherent ambiguity in occluded regions caused by insufficient geometric constraints. To address these issues, this paper presents a reformulated benchmark for unsupervised monocular 3D occupancy prediction. We first interpret the variables involved in the volume rendering process and identify the most physically consistent representation of the occupancy probability. Building on these analyses, we improve existing evaluation protocols by aligning the newly identified representation with voxel-wise 3D occupancy ground truth, thereby enabling unsupervised methods to be evaluated in a manner consistent with that of supervised approaches. Additionally, to impose explicit constraints in occluded regions, we introduce an occlusion-aware polarization mechanism that incorporates multi-view visual cues to enhance discrimination between occupied and free spaces in these regions. Extensive experiments demonstrate that our approach not only significantly outperforms existing unsupervised approaches but also matches the performance of supervised ones. Our source code and evaluation protocol will be made available upon publication.

## 1 Introduction

3D occupancy prediction, which infers the volumetric structure of real-world environments, enables unified spatial representations that support various downstream tasks in autonomous systems (Cao et al., 2022; Huang et al., 2023; Li et al., 2023). Most existing methods (Zhang et al., 2023; Jiang et al., 2024) rely on supervised learning with voxel-wise annotated 3D ground truth, typically generated from sparse LiDAR point clouds (Tian et al., 2023; Wei et al., 2023). Acquiring such annotations is, nevertheless, both labor-intensive and prone to inaccuracies, thereby impeding large-scale training. In contrast, unsupervised methods (Wimbauer et al., 2023; Han et al., 2024; Li et al., 2024a; Feng et al., 2025) based on neural radiance fields (NeRFs) (Mildenhall et al., 2020) avoid the need for explicit supervisory signals and realize occupancy inference from a single image, demonstrating strong potential and flexibility for real-world applications.

As 3D occupancy prediction continues to advance, the systematic evaluation of emerging networks has become increasingly critical (Zhang et al., 2024). While supervised methods can be evaluated on well-established benchmarks (Wang et al., 2025; Tian et al., 2023; Wei et al., 2023; Wang et al., 2023), unsupervised NeRF-based approaches, developed since BTS (Wimbauer et al., 2023), are still evaluated using inappropriate protocols misaligned with the 3D nature of the task. Specifically, NeRF networks are designed to output implicit rendering weights for alpha compositing. As pointed out by Ahn et al. (2024), the magnitude of these weights depends on the scale of the sampling interval. However, existing evaluation protocols erroneously equate these scale-variant, point-wise weights with fixed-range, voxel-wise occupancy ground truth, thereby introducing inconsistencies between the training and evaluation protocols. In addition, existing occupancy annotations are technically limited to a 2D plane, which is ill-suited for an inherently 3D task, thereby undermining both the reliability and completeness of the evaluation results.

The aforementioned issues in current evaluation protocols obscure the inherent limitations of existing methods. Following the NeRF paradigm, early representative monocular approaches (Wimbauer et al., 2023; Han et al., 2024; Li et al., 2024a; Feng et al., 2025) reconstruct target-view images from multiple source views through volume rendering. These networks are trained by minimizing the photometric discrepancies between reconstructed and real images. However, during the volume rendering process, density values in occluded regions contribute minimally to the reconstructed image, as image intensities from these areas are rarely transmitted through foreground occluders during rendering integration for the target view. Compared to supervised approaches, which can directly learn from occluded occupancy ground truth, NeRF-based networks inherently struggle to accurately model occupancy distributions in these regions with only 2D supervision. When the ground truth dimension is lifted to 3D, the accuracy of existing methods deteriorates greatly due to the increased proportion of occluded regions.

Therefore, in this study, we rebenchmark the unsupervised monocular 3D occupancy prediction task to address all the aforementioned challenges. First, we systematically analyze and interpret the occupancy probability in NeRF-based methods, and incorporate spatial neighborhood into point-wise occupancy estimations. This integration mitigates the magnitude variations of network outputs and alleviates spatial misalignment with voxel-wise ground truth. Furthermore, we transform the original camera coordinate system into a new space and develop an occupancy sampling algorithm to align the spatial distribution of the proposed occupancy representation with that of the 3D occupancy annotations. This algorithm enables a reliable and interpretable benchmark aligned with the standard 3D evaluation protocols widely used for supervised methods (Li et al., 2024b). Moreover, we design an occlusion-aware occupancy polarization mechanism by correlating image intensity variations with occupancy discrepancies across multiple views to provide additional supervisory signals for occluded regions. Extensive experimental results on the KITTI-360 (Liao et al., 2022) dataset validate both the interpretability and rationality of our reformulated benchmark, as well as the effectiveness of the proposed occupancy polarization mechanism. In addition, comprehensive comparisons with supervised methods underscore the state-of-the-art (SoTA) performance achieved by our unsupervised approach. In a nutshell, the key contributions of this study are as follows:

- We delve into the interpretation of occupancy probability in NeRF, bridging the gap between NeRF-based predictions and voxel-wise 3D occupancy evaluation protocols.

- We develop a coordinated-transformed sampling algorithm that unifies the benchmark for both unsupervised and supervised 3D occupancy prediction approaches.

- We propose an occlusion-aware occupancy polarization mechanism that exploits visual cues from other views to provide additional supervision in occluded areas.

## 2 RELATED WORK

### 2.1 SUPERVISED 3D OCCUPANCY PREDICTION

Learning voxel-wise 3D occupancy from images is a key step toward comprehensive 3D scene understanding (Wang et al., 2024; Ma et al., 2024; Li et al., 2025). As a pioneering study, MonoScene (Cao et al., 2022) introduces a 3D occupancy prediction framework that infers voxel-level geometry and semantics from a single image. TPVFormer (Huang et al., 2023) extends this approach to multi-camera settings by incorporating tri-perspective representations. Subsequent studies have progressively refined network architectures within this end-to-end framework. For instance, VoxFormer (Li et al., 2023) adopts Transformers over sparse voxels for long-range context modeling, while OccFormer (Zhang et al., 2023) leverages a dual-path Transformer architecture to fuse semantic and geometric features across multiple views. Building upon these prior works, Symphonies (Jiang et al., 2024) leverages contextual instance queries to enhance scene-level geometric and semantic understanding in complex driving scenes. Beyond these end-to-end frameworks that directly infer voxel-level occupancies, recent methods have explored implicit representations to improve both accuracy and interpretability. HybridOcc (Zhao et al., 2024) bridges explicit and implicit representations by integrating NeRF branches with Transformer-based voxel queries, which leads to significantly improved performance. Nevertheless, these methods rely on the costly process of acquiring accurate 3D annotations, which limits their scalability for large-scale training. Thus, this study focuses extensively on unsupervised methods.

## 2.2 Unsupervised 3D Occupancy Prediction

Unsupervised methods aim to reconstruct 3D scene geometry with only 2D supervision (Huang et al., 2024a; Jevtić et al., 2025). Most existing methods are built upon NeRFs (Mildenhall et al., 2020), which utilize a continuous volume rendering mechanism and optimize the network by minimizing the photometric loss across multiple views. BTS (Wimbauer et al., 2023) is a pioneering work that presents a fully unsupervised NeRF pipeline for single-view 3D reconstruction through differentiable volume rendering. Building upon this foundation, KDBTS (Han et al., 2024) distills multi-view density fields into a single-view network via self-supervised training, thereby greatly improving its performance across diverse scenes. Subsequent studies (Li et al., 2024a; Feng et al., 2025) have increasingly incorporated off-the-shelf vision models to improve object-level 3D occupancy predictions. For instance, KYN (Li et al., 2024a) leverages vision-language priors to integrate semantic knowledge and spatial context into the pipeline for semantically guided 3D geometric reasoning. ViPOcc (Feng et al., 2025) further introduces visual priors from foundation models to enhance instance-level semantic reasoning and temporal photometric consistency. Despite these advances, existing approaches remain constrained by their reliance primarily on a reconstruction loss through volume rendering, which inherently fails to provide explicit guidance in occluded regions. Additionally, they often overlook the inconsistency between training and evaluation protocols, ultimately compromising the reliability of 3D occupancy predictions. In this work, we present a more interpretable representation of occupancy probability and propose an occlusion-aware polarization mechanism to solve these issues.

## 2.3 Benchmarks for 3D Occupancy Prediction

Several datasets (Liao et al., 2022; Caesar et al., 2020) provide video sequences accompanied by camera poses and LiDAR point clouds collected in real-world driving environments. To enable 3D occupancy prediction evaluation, recent studies (Wei et al., 2023; Li et al., 2024b) have constructed voxel-level occupancy annotations by aggregating multi-frame LiDAR point clouds. Occ3D (Tian et al., 2023) is among the first to achieve voxel-level semantic annotations, enabling dense 3D occupancy evaluation at fine granularity. SurroundOcc (Wei et al., 2023) applies Poisson reconstruction to consolidate LiDAR scans into dense 3D annotations, while OpenOccupancy (Wang et al., 2023) improves labeling accuracy through extensive manual annotation to mitigate LiDAR sparsity. Recent benchmarks such as SSCBench (Li et al., 2024b) and UniOcc (Wang et al., 2025) extend unified evaluation protocols to a variety of driving scenes. Despite recent progress focused primarily on supporting supervised learning paradigms, a standardized benchmark for unsupervised 3D occupancy learning remains underdeveloped, with existing annotations often limited to a single 2D plane. In this work, we exclusively utilize the aforementioned voxel-level occupancy annotations to evaluate unsupervised approaches, thereby establishing a comprehensive 3D benchmarking protocol.

## 3 Methodology

### 3.1 Problem Setup

In NeRF-based approaches, the network with parameters $\boldsymbol{\Theta}$ takes as input a target-view RGB image $\boldsymbol{I}_0$, camera intrinsic matrix $\boldsymbol{K}$, and a 3D point $\boldsymbol{x}^{(i)}$ to predict a rendering weight $\sigma^{(i)}$ as follows:

$$\sigma^{(i)} = f\left(\boldsymbol{I}_0, \boldsymbol{K}, \boldsymbol{x}^{(i)}; \boldsymbol{\Theta}\right),\tag{1}$$

where the 3D point $\boldsymbol{x}^{(i)} = \boldsymbol{o} + t^{(i)}\boldsymbol{d}$ is sampled along a ray in the direction of the unit vector $\boldsymbol{d}$, with $t^{(i)}$ denoting the distance from the camera origin $\boldsymbol{o}$ to $\boldsymbol{x}^{(i)}$. During the volume rendering process, the opacity $\alpha^{(i)}$ is first computed along the ray using the following expression:

$$\alpha^{(i)} = 1 - \exp\left(-\sigma^{(i)}\delta^{(i)}\right),\tag{2}$$

where $\delta^{(i)} = |\boldsymbol{x}^{(i+1)} - \boldsymbol{x}^{(i)}|$ denotes the length of the ray segment between consecutive sample points $\boldsymbol{x}^{(i)}$ and $\boldsymbol{x}^{(i+1)}$. An image in the target view is rendered using the following expression:

$$\hat{\boldsymbol{c}} = \sum_{i=1}^{N} \alpha^{(i)} T^{(i)} \boldsymbol{c}^{(i)}, \quad T^{(i)} = \prod_{j=1}^{i-1}\left(1 - \alpha^{(j)}\right),\tag{3}$$

where $\hat{c}$ denotes the image intensity rendered along a sampled ray with $N$ sampled points, $c^{(i)}$ denotes the color at point $x^{(i)}$ sampled from other viewpoints, and $T^{(i)}$ represents the accumulated transmittance up to the $i$-th point. The network is trained by minimizing a photometric loss that quantifies the discrepancy between the rendered and ground-truth image intensities.

### 3.2  OCCUPANCY PROBABILITY INTERPRETATION FOR NERF

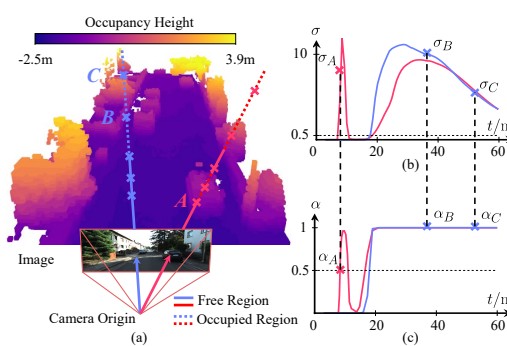

Figure 1: A comparison between the network output $\sigma$ and the opacity $\alpha$ during inference: (a) two representative sampled rays; (b) $\sigma$ distributions; (c) $\alpha$ distributions. For point A, which transitions from occupied to free space, $\alpha_A$ is bounded within the range $(0, 1)$, whereas $\sigma_A$ has no upper bound, making $\alpha$ a more suitable representation for occupancy probability; For points B and C with identical occupancy status, their discrepancy in $\sigma$ is significantly greater than that in $\alpha$, demonstrating that our proposed representation for occupancy probability effectively eliminates the magnitude variation caused by non-uniform point sampling.

Existing evaluation protocols simply adopt a fixed $\sigma$ threshold of 0.5 to binarize each voxel[1]:

$$o(x) = [\sigma > 0.5], \qquad (4)$$

where $[\cdot]$ denotes the Iverson bracket, and $x$ represents the voxel center. However, according to Eq. 1, the network predicts occupancy densities at infinitesimal points, whereas the ground truth typically corresponds to the occupied probability of a volumetric cell. The spatial misalignment between point-wise predictions and voxel-level ground truth annotations makes their direct comparison with a fixed threshold of 0.5 in Eq. 4 uninterpretable.

Specifically, existing approaches typically perform non-uniform point sampling along each ray during training, with denser sampling near the camera and sparser sampling in more distant regions. The variation in sampling density along the ray changes the spatial neighborhood around each point, which is characterized by the distance $\delta$ to its nearest point. Consider two occupied points, B and C, as illustrated in Fig. 1(a), with the former located in a densely sampled region and the latter located in a sparsely sampled one. According to Eq. 3, the rendering contributions $\alpha T$ of these two occupied points to the reconstructed pixel color are theoretically expected to be equivalent. In the training process, the network adjusts its output $\sigma$ to satisfy this equality, which leads to a magnitude variation issue, as illustrated in Fig. 1(b). As indicated by Eq. 2, denser sampling yields a smaller spatial neighborhood $\delta_B$, which in turn increases the network output $\sigma_B$, and conversely, a larger $\delta_C$ in sparse sampling regions results in a lower $\sigma_C$. This dependency established during training is retained at inference time, causing magnitude variations of the network outputs along the ray. Current method, nevertheless, overlooks this variation and applies a fixed threshold to obtain voxel-wise occupancy predictions, leading to inconsistencies in existing evaluation protocols across regions with varying point sampling densities. In addition, when the number of sampled points along each ray changes, the spatial neighborhood $\delta$ around each point also varies, thereby inducing inconsistent output magnitudes due to the underlying dependency. This also leads to inconsistency of evaluation protocols across different experimental configurations.

In this study, we argue that the opacity $\alpha$ provides a more interpretable representation of occupancy probability than the network's output $\sigma$. According to Eq. 2, the opacity, computed based on the spatial neighborhood $\delta$, characterizes the occupancy attributes within a finite volume rather than at an infinitesimal point. This volumetric interpretation aligns more naturally with voxel-level occupancy ground truth, making it better suited for model evaluation. In addition, unlike the original network output $\sigma$, which is highly sensitive to the point sampling strategy, the opacity value is bounded within $(0, 1)$, as illustrated in Fig. 1(c). This bounded range eliminates the aforementioned magnitude variation effect, thereby enhancing the consistency of the evaluation protocols across diverse

---

[1]In this subsection, we omit the superscript $^{(i)}$ for $\sigma$, $\alpha$, $\delta$, and $T$ for notational simplicity.

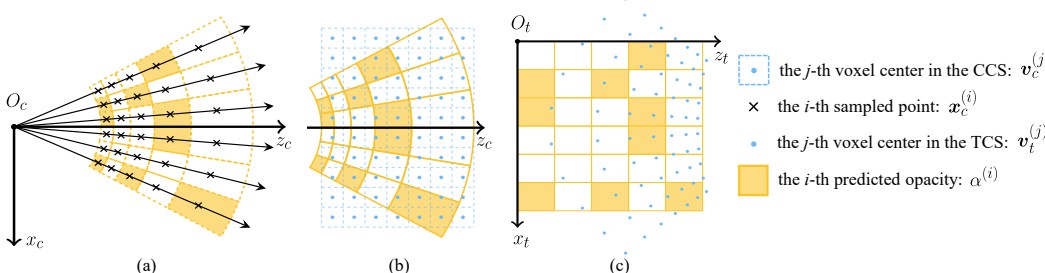

the $j$-th voxel center in the CCS: $\boldsymbol{v}_c^{(j)}$

$\times$ the $i$-th sampled point: $\boldsymbol{x}_c^{(i)}$

the $j$-th voxel center in the TCS: $\boldsymbol{v}_t^{(j)}$

the $i$-th predicted opacity: $\alpha^{(i)}$

(a)      (b)      (c)

Figure 2: The occupancy sampling algorithm in the camera coordinate system (CCS) and the transformed coordinate system (TCS): (a) network inference with sampled points as input; (b) opacity distribution v.s. the voxel grid in the CCS; (c) opacity sampling using voxel centers in the TCS.

settings. The occupancy prediction is formulated under the proposed interpretation as follows:

$$o(\boldsymbol{x}) = [\alpha > 0.5]. \tag{5}$$

By adopting this opacity-based volumetric interpretation, we redefine the occupancy probability representation and reformulate the entire benchmark.

### 3.3 COORDINATE-TRANSFORMED OCCUPANCY SAMPLING

As discussed in the previous subsection, opacity $\alpha$ provides a more appropriate representation for occupancy prediction. Nonetheless, as illustrated in Figs. 2(a) and (b), the predicted opacities are distributed along radial segments originating from the camera center, whereas the ground-truth occupancy annotations are defined on a uniform voxel grid. To address this spatial misalignment problem, we propose a coordinate-transformed occupancy sampling algorithm that maps opacities from radial segments onto the voxel grid.

As illustrated in Fig. 2(c), to more clearly characterize the radial distribution of opacity, we construct the TCS, in which opacity is uniformly distributed, from the CCS. A 3D point $\boldsymbol{x}_c = (x_c, y_c, z_c)^\top$ in the CCS corresponds to the homogeneous pixel coordinates $(u, v, 1)^\top = \boldsymbol{K}\boldsymbol{x}_c/z_c$, where $\boldsymbol{K}$ denotes the camera intrinsic matrix. The coordinates of the corresponding point in the TCS can be computed using the following expression:

$$\boldsymbol{x}_t = \left( \frac{u}{w-1}, \frac{v}{h-1}, \frac{1/t_n - 1/\|\boldsymbol{x}_c\|_2}{1/t_n - 1/t_f} \right)^\top, \tag{6}$$

where the image resolution is $h \times w$ pixels, and $t_n$ and $t_f$ denote the near and far bounds, respectively. This transformation maps the view frustum in the rendering field into a normalized cube spanning $[0, 1]^3$, where the $x$- and $y$-axes in the TCS are aligned with the image axes, and the $z$-axis is aligned with the sampled ray direction. The TCS enables the grid sampling process, which requires a uniformly distributed opacity map.

Given the above details on the defined TCS, the coordinate-transformed occupancy sampling algorithm proceeds with the following steps. First, each 3D point $\boldsymbol{x}_c^{(i)}$ is sampled in the CCS using the same strategy adopted during training. Following Eq. 1, $\boldsymbol{x}_c^{(i)}$ is passed through the network to infer the cooresponding output $\sigma^{(i)}$, as depicted in Fig. 2(a). Following Eq. 2, the opacity $\alpha^{(i)}$ is then computed along each camera ray using the sampling interval $\delta^{(i)}$, as shown in Fig. 2(b). Subsequently, each voxel center $\boldsymbol{v}_c^{(j)}$ is transformed from the CCS to the TCS, yielding the corresponding coordinates $\boldsymbol{v}_t^{(j)}$, as illustrated in Fig. 2(c). The voxel-wise occupancy predictions, spatially aligned with the ground truth annotations, are obtained by sampling the grid-based opacity map in the TCS, expressed as follows:

$$o(\boldsymbol{v}_c^{(j)}) = \left[ \mathcal{A}\langle\boldsymbol{v}_t^{(j)}\rangle > 0.5 \right], \tag{7}$$

where $\mathcal{A}\langle\cdot\rangle$ denotes the grid sampling process on the opacity map $\mathcal{A} = \{\alpha^{(i)}\}$. The resulting occupancy predictions are subsequently evaluated using the metrics in our benchmark.

### 3.4 OCCLUSION-AWARE OCCUPANCY POLARIZATION

Further exploration of our benchmark reveals a critical limitation of existing unsupervised NeRF-based occupancy prediction methods: they inherently struggle to predict occupancy distributions behind foreground occluders. This limitation arises from their exclusive reliance on reconstruction loss derived from volume rendering. To better understand this limitation, we conduct a quantitative analysis on the supervisory signals within occluded regions during the back-propagation process. For simplicity, we define the per-pixel photometric reconstruction loss as $\mathcal{L}_r = |\hat{c} - c_{gt}|$, where $c_{gt}$ denotes the ground-truth RGB value. Taking the derivative of Eq. 3 results in the gradient of the loss with respect to the predicted opacity at the corresponding sampled point $x^{(i)}$ as follows:

$$\frac{\partial \mathcal{L}_r}{\partial \alpha^{(i)}} = T^{(i)} [\hat{c} > c_{gt}]^\top c^{(i)} - T^{(i-1)} \sum_{j=i+1}^{N} \left( \alpha^{(i)} \prod_{k=i+1}^{j-1} (1 - \alpha^{(k)}) [\hat{c} > c_{gt}]^\top c^{(j)} \right) \quad (8)$$

The detailed derivation of Eq. 8 is given in the supplement. As shown in Eq. 3, the transmittance $T^{(i)}$ decreases monotonically as the depth of $x^{(i)}$ increases, approaching zero in regions occluded by foreground occluders. According to Eq. 8, when both $T^{(i-1)}$ and $T^{(i)}$ approach zero, the gradient $\partial L_r / \partial \alpha^{(i)}$ diminishes as well. As a result, the gradients with respect to the network parameters in occluded regions become negligible, thereby hindering effective learning in these areas.

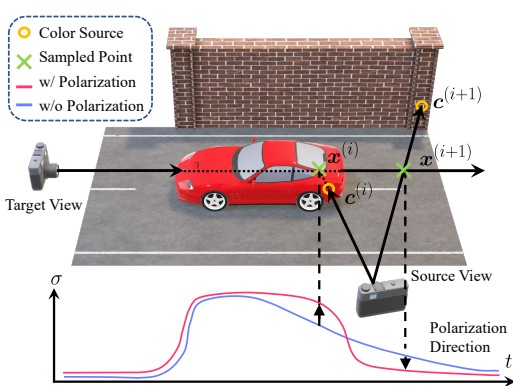

Figure 3: An illustration of the occlusion-aware occupancy polarization mechanism. The discrepancy of sampled colors on adjacent sampled points indicates that the colors likely originate from distinct objects. The mechanism amplifies the occupancy differences between such points and refines predictions in occluded regions.

To address this issue, we leverage visual cues from other views to incorporate additional, explicit supervisory signals. As illustrated in Fig. 3, although occupancy in occluded regions is invisible in the target view, it may become visible in certain source views. Consider two adjacent sampled points, $x^{(i)}$ and $x^{(i+1)}$, located along a single ray. If at least one of them, e.g., $x^{(i)}$, is occupied, the color difference between the two points provides valuable visual cues about the underlying occupancy. When the neighboring point $x^{(i+1)}$ is unoccupied, its sampled color $c^{(i+1)}$ often originates from a surface different from that of $x^{(i)}$, leading to a noticeable color discrepancy. In contrast, if both points are occupied and lie on the same object, their colors tend to be similar due to their shared surface. Thus, the color discrepancy or similarity between adjacent points serves as an effective indicator of local occupancy variation. Nonetheless, when both points are unoccupied, the observed color difference is typically unrelated to underlying scene geometry and provides limited value for occupancy learning.

Motivated by the observation, we develop an occlusion-aware occupancy polarization mechanism to explicitly guide occupancy predictions across occupied and free space, enhancing the supervision signals in occluded regions. Specifically, this mechanism encourages the network to polarize the predicted occupancy values of adjacent points $x^{(i)}$ and $x^{(i+1)}$ when their sampled colors differ significantly. This facilitates sharper occupancy transitions in regions where visual cues suggest a boundary. We implement this mechanism by formulating a polarization loss $\mathcal{L}_p$ as follows:

$$\mathcal{L}_p = \sum_{i=1}^{N-1} M_i \left| c^{(i+1)} - c^{(i)} \right| \exp \left( - \left| \sigma^{(i+1)} - \sigma^{(i)} \right| \right), \quad (9)$$

where $M_i = \max(\alpha^{(i)}, \alpha^{(i+1)})$ is a weighting mask to exclude regions where both consecutive points are unoccupied, as discussed above. The loss penalizes insufficient polarization across object boundaries and diminishes as the predicted occupancy difference increases. The overall loss is:

$$\mathcal{L} = \lambda_r \mathcal{L}_r + \lambda_p \mathcal{L}_p \quad (10)$$

where $\lambda_r$ and $\lambda_p$ are the weighting parameters.

Table 1: Quantitative comparison among unsupervised methods on the KITTI-360 dataset.

| Method | $O_{Acc}$ | $O_{Pre}$ | $O_{Rec}$ | $IE_{Acc}$ | $IE_{Pre}$ | $IE_{Rec}$ |
|---|---|---|---|---|---|---|
| BTS (Wimbauer et al., 2023) | 0.870 | 0.733 | 0.745 | 0.727 | 0.466 | 0.658 |
| KDBTS (Han et al., 2024) | 0.871 | 0.746 | 0.731 | 0.722 | 0.463 | 0.682 |
| KYN (Li et al., 2024a) | 0.861 | 0.746 | 0.654 | 0.671 | 0.402 | **0.707** |
| ViPOcc (Feng et al., 2025) | 0.875 | 0.748 | 0.746 | 0.728 | 0.467 | 0.668 |
| **Ours** | **0.883** | **0.763** | **0.757** | **0.741** | **0.475** | 0.676 |

## 4 EXPERIMENTS

### 4.1 IMPLEMENTATION DETAILS

**Datasets.** We establish a new benchmark and conduct extensive experiments on the KITTI-360 dataset (Liao et al., 2022), with 3D occupancy ground truth provided by the SSCBench-KITTI-360 dataset (Li et al., 2024b). Unsupervised methods are trained with video sequences and the corresponding ground-truth camera poses from the KITTI-360 dataset. All images are resized to a resolution of $192 \times 640$ pixels. Following Wimbauer et al. (2023), we split the dataset into a training set of 98,008 images, a validation set of 11,451 images, and a test set of 446 images.

**Network Training.** We train our network (He et al., 2016) for 25 epochs using the Adam (Kingma et al., 2014) optimizer on an NVIDIA RTX 4090 GPU, with the initial learning rate set to $2 \times 10^{-4}$, which is decayed by a factor of 2 during the final 10 epochs.

**Evaluation Protocols.** Existing evaluation protocols for unsupervised approaches are typically restricted to a narrow 2D slice of the scene, with highly limited spatial ranges ($y = 0.375$m, $x \in [-4\text{m}, +4\text{m}]$, and $z \in [3\text{m}, 20\text{m}]$ in the CCS). To overcome this limitation, we utilize the 3D occupancy ground truth from SSCBench-KITTI-360 dataset (Li et al., 2024b), which covers a substantially larger spatial volume extending 51.2m forward, 25.6m to each side, and 6.4m in height, discretized into a $256 \times 256 \times 32$ voxel grid with a resolution of 0.2m. Specifically, we align the 3D occupancy annotations for each image in the KITTI-360 test set and provide the transformation from the voxel coordinate system to the camera coordinate system. With this transformation, we generate 3D frustum masks and 3D voxel visibility masks using a ray-tracing algorithm, thereby enabling evaluation in occluded regions. We relax the evaluation limitation along the $y$-axis while preserving the original ranges along the $x$- and $z$-axes in Wimbauer et al. (2023). This adaptation enables the evaluation protocol to prioritize spatial regions closer to the input camera viewpoint, where predictions are generally more reliable. Additional details are provided in the supplement.

**Metrics.** Based on the above evaluation protocols, we extend the evaluation metrics used in Wimbauer et al. (2023) to the 3D domain. These metrics include: occupancy accuracy $O_{Acc}$, occupancy precision $O_{Pre}$, occupancy recall $O_{Rec}$, invisible and empty accuracy $IE_{Acc}$, invisible and empty precision $IE_{Pre}$, and invisible and empty recall $IE_{Rec}$. The first three metrics are computed within the camera frustum, while the latter three are evaluated within the intersection of the camera frustum and the invisibility mask. In addition, to facilitate comparison with supervised methods, we adopt unified metrics, including intersection over union (IoU), precision (Pre), and recall (Rec).

### 4.2 COMPARISONS WITH STATE-OF-THE-ART METHODS

We compare our approach with representative unsupervised SoTA methods. As shown in Table 1, our method achieves SoTA performance across the majority of evaluation metrics. In particular, it improves $O_{Acc}$, $O_{Pre}$, $O_{Rec}$, $IE_{Acc}$, and $IE_{Pre}$ by 0.9%, 2.0%, 1.5%, 1.8%, and 1.7%, respectively. It is worth noting that although KYN achieves the highest $IE_{Rec}$ score, it incorporates a computation-ally intensive visual-language network (Li et al., 2022), which significantly compromises inference efficiency. Qualitative comparisons are presented in Fig. 4, where the predicted occupancy grids are visualized from the right side of the scene. Compared to BTS and ViPOcc, our method achieves superior 3D geometry reconstruction, and effectively mitigates trailing effects. The results demonstrate the effectiveness of our proposed polarization mechanism in reasoning occluded occupancy.

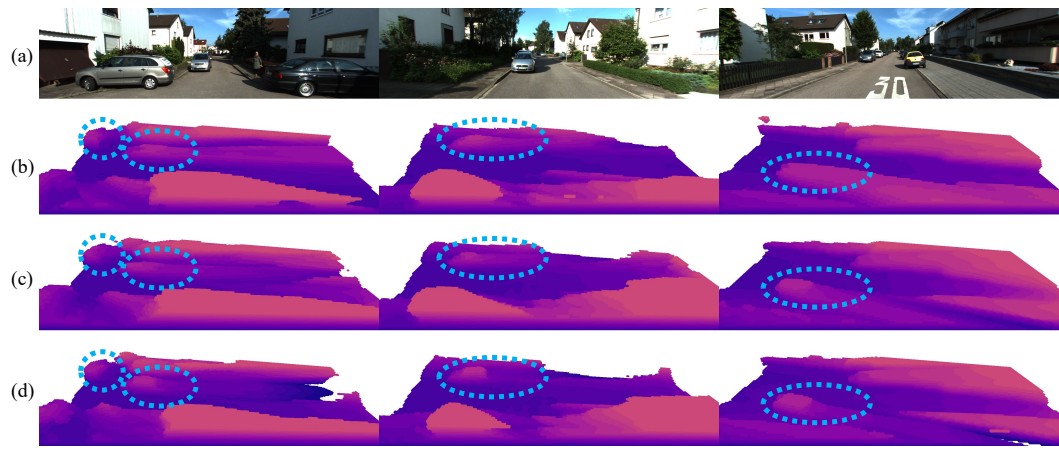

Figure 4: Qualitative comparisons of 3D occupancy prediction on the KITTI-360 dataset: (a) input RGB images; (b) BTS results; (c) ViPOcc results; (d) our results.

Table 2: Quantitative comparison with both supervised and unsupervised methods on the KITTI-360 dataset. The best results are shown in bold type, with the best unsupervised ones underlined.

| Supervised Method | IoU (%) | Pre (%) | Rec (%) | Unsupervised Method | IoU (%) | Pre (%) | Rec (%) |
|---|---|---|---|---|---|---|---|
| MonoScene | 37.9 | 56.7 | 53.3 | KDBTS | 44.6 | 52.7 | 74.3 |
| VoxFormer | 38.8 | 58.5 | 53.4 | KYN | 44.4 | 54.0 | 71.4 |
| OccFormer | 40.3 | 59.7 | 55.3 | ViPOcc | 43.1 | 47.2 | **83.4** |
| Symphonies | 44.1 | **69.2** | 54.9 | **Ours** | **45.5** | 50.8 | 81.4 |

Additionally, our reformulated benchmark suite is aligned with the evaluation protocols used by supervised methods, thereby enabling direct comparison with representative supervised approaches. The quantitative experimental results presented in Table 2 reveal several noteworthy findings. Most notably and somewhat unexpectedly, NeRF-based methods achieve IoU scores that are comparable to, or even exceed those of recent supervised approaches, while outperforming most earlier ones. We attribute this phenomenon to the limited quality of existing 3D occupancy ground truth, which may introduce misleading supervisory signals and thus

Table 3: Zero-shot 3D occupancy prediction metrics on the SemanticKITTI dataset.

| Method | IoU (%) | Pre (%) | Rec (%) |
|---|---|---|---|
| SceneRF | 13.8 | 17.3 | 41.0 |
| SelfOcc(BEV) | 21.0 | **37.3** | 32.4 |
| SelfOcc(TPV) | 22.0 | 34.8 | 37.3 |
| ViPOcc | 23.6 | 26.9 | 66.8 |
| Ours | **24.1** | 27.0 | **68.7** |

hinder the effectiveness of supervised training. In contrast, NeRF-based methods are unaffected by this issue, as they do not rely on such supervision. Moreover, unsupervised methods typically exhibit higher recall than precision, indicating a tendency to overestimate occupied space, particularly in occluded regions where supervisory signals are absent. In contrast, benefiting from direct supervision in these areas, supervised methods often achieve more balanced metrics. While our method mitigates this imbalance compared to the unsupervised baseline ViPOcc, further improvements are necessary, especially in handling occlusions, which remain a key challenge for unsupervised 3D occupancy prediction.

To further evaluate the generalizability of the proposed method, we conduct a zero-shot test on the SemanticKITTI dataset (Behley et al., 2019) using the pre-trained weights obtained from the KITTI-360 dataset. As presented in Table 3, the proposed method outperforms other SoTA methods including SceneRF (Cao & De Charette, 2023) and SelfOcc (Huang et al., 2024b) in zero-shot 3D occupancy prediction, demonstrating its exceptional generalizability.

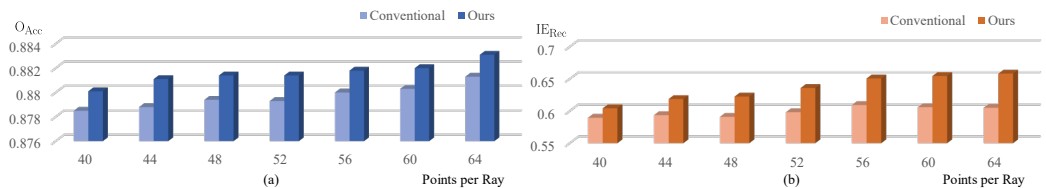

Figure 5: Comparison between the conventional and proposed occupancy probability representations in terms of (a) $O_{Acc}$ and (b) $IE_{Rec}$ across varying numbers of sampled points per ray.

## 4.3 ABLATION STUDIES

**Occupancy Probability Interpretation.** To demonstrate the rationality of our interpreted occupancy probability representation, we evaluate existing unsupervised methods under both the conventional and the proposed representations. Specifically, we utilize pretrained weights from prior works without any modification and exclusively adjust the occupancy probability representation for performance evaluation. The network's performance is then evaluated using the above-mentioned metrics under both interpretations for comparative analysis.

As shown in Table 4, directly applying our interpreted occupancy probability defined in Eq. 7, without any retraining, consistently leads to higher $O_{Acc}$ and $IE_{Rec}$ scores, compared to the conventional representation defined in Eq. 4. This improvement demonstrates that our proposed formulation ensures greater consistency between the training and evaluation protocols and is more suitable for the quantitative evaluation of unsupervised methods. Furthermore, by leveraging the proposed occupancy probability representation, we observe opposing trends in $IE_{Rec}$ and $IE_{Acc}$, which quantify performance within invisible regions, revealing that existing methods generally misclassify free space as occupied when explicit supervisory signals are absent. This observation corroborates the limitation in current approaches when inferring occupancy in occluded regions.

To further demonstrate the robustness of the proposed occupancy probability representation under varying point sampling intervals along rays, we train the baseline network (Feng et al., 2025) with different numbers of sampled points per ray, while maintaining fixed near and far bounds, as defined in the NeRF framework. As illustrated in Fig. 5, unlike the conventional representation, which suffers from fluctuations in network output magnitude under varying sampling densities, our representation maintains consistent performance, demonstrating greater stability to changes in point sampling strategies, as discussed above.

Table 4: Ablation study on the occupancy probability interpretation. The conventional occupancy probability interpretation is given in Eq. 4, whereas ours is given in Eq. 7.

| Method | Representation | $O_{Acc}$ | $IE_{Acc}$ | $IE_{Rec}$ |
|---|---|---|---|---|
| BTS | Conventional | 0.867 | **0.756** | 0.606 |
| | Ours | **0.870** | 0.727 | **0.658** |
| KDBTS | Conventional | 0.868 | **0.750** | 0.618 |
| | Ours | **0.871** | 0.722 | **0.682** |
| ViPOcc | Conventional | 0.873 | **0.757** | 0.608 |
| | Ours | **0.875** | 0.728 | **0.668** |

Table 5: Ablation study on the occlusion-aware occupancy polarization mechanism.

| Baseline | $\mathcal{L}_p$ | $O_{Acc}$ | $IE_{Acc}$ | $IE_{Rec}$ |
|---|---|---|---|---|
| BTS | ✗ | 0.870 | 0.727 | 0.658 |
| | ✓ | **0.880** | **0.737** | **0.667** |
| KDBTS | ✗ | 0.871 | 0.722 | **0.682** |
| | ✓ | **0.879** | **0.725** | **0.682** |
| ViPOcc | ✗ | 0.875 | 0.728 | 0.668 |
| | ✓ | **0.883** | **0.741** | **0.676** |

**Occlusion-Aware Occupancy Polarization.** To validate the effectiveness of the occlusion-aware occupancy polarization mechanism, we incorporate its corresponding loss $\mathcal{L}_p$ into the overall loss function and retrain several baseline networks for comprehensive comparisons. As shown in Table 5, the mechanism consistently improves all evaluation metrics across all baseline networks, with maximum improvements of 1.1%, 1.8% and 1.4% on $O_{Acc}$, $IE_{Acc}$ and $IE_{Rec}$, demonstrating its general efficacy. Additional comparative results are provided in the supplement.

## 5 CONCLUSION

In this paper, we first addressed a critical limitation in the existing unsupervised monocular 3D occupancy prediction benchmark: the spatial inconsistency between training and evaluation protocols. To this end, we developed an interpretable, opacity-based representation of occupancy probability and introduced a coordinate-transformed sampling algorithm for voxel-wise occupancy prediction, contributing a consistent and reliable evaluation protocol aligned with those used by supervised methods. In addition, to compensate for the inherent lack of photometric supervision revealed by the proposed benchmark, we leveraged multi-view visual cues and introduced an occlusion-aware occupancy polarization mechanism, which proves to be compatible across all baseline networks. Extensive experiments conducted with both supervised and unsupervised methods on the reformulated benchmark validate the rationality of our interpreted occupancy probability, the alignment between training and evaluation protocols, and the effectiveness of the proposed occlusion-aware occupancy polarization mechanism.

## REPRODUCIBILITY STATEMENT

Our results are reproducible by consulting section 4 and section C in appendix for the experimental details of our new benchmark, and by downloading the supplementary materials which contain the complete source code.

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

## A  ETHICS

In this study, we utilize the KITTI-360 (Liao et al., 2022) and SSCBench-KITTI-360 (Li et al., 2024b) datasets for the development and evaluation of 3D occupancy prediction networks. It is important to emphasize that we strictly adhere to the terms of usage for each dataset. We acknowledge that these datasets may contain images with visible faces and other personal data collected without consent. Nevertheless, we confirm that no biometric information has been processed. All images are used in accordance with the CC-BY license or in a manner compatible with the Data Analysis Permission.

## B  ADDITIONAL METHODOLOGICAL DETAILS

### B.1  THE GRADIENT OF THE RECONSTRUCTION LOSS

We derive the gradient of the reconstruction loss with respect to the opacity at the sampled point $\boldsymbol{x}^{(i)}$ as follows:

$$\frac{\partial \mathcal{L}_r}{\partial \alpha^{(i)}} = \frac{\partial \mathcal{L}_r}{\partial \hat{\boldsymbol{c}}} \frac{\partial \hat{\boldsymbol{c}}}{\partial \alpha^{(i)}} = [\hat{\boldsymbol{c}} > \boldsymbol{c}_{gt}]^\top \frac{\partial \hat{\boldsymbol{c}}}{\partial \alpha^{(i)}}, \tag{11}$$

where $[\cdot]$ denotes the Iverson bracket, $\mathcal{L}_r$ represents the reconstruction loss, $\alpha^{(i)}$ denotes the opacity at point $\boldsymbol{x}^{(i)}$, $\hat{\boldsymbol{c}} = \sum_{i=1}^{N} \alpha^{(i)} T^{(i)} \boldsymbol{c}^{(i)}$ represents the rendered pixel color, and $\boldsymbol{c}_{gt}$ denotes the ground-truth RGB value. We take the derivate of the rendered pixel color $\hat{\boldsymbol{c}}$ and obtain the following expression:

$$\frac{\partial \hat{\boldsymbol{c}}}{\partial \alpha^{(i)}} = \sum_{j=1}^{N} \frac{\partial}{\partial \alpha^{(i)}} \left( \alpha^{(j)} T^{(j)} \boldsymbol{c}^{(j)} \right)$$
$$= T^{(i)} \boldsymbol{c}^{(i)} + \sum_{j=1}^{N} \alpha^{(j)} \boldsymbol{c}^{(j)} \frac{\partial T^{(j)}}{\partial \alpha^{(i)}}, \tag{12}$$

where $\partial T^{(j)}/\partial \alpha^{(i)}$ is obtained by taking the derivative of $T^{(j)} = \prod_{k=1}^{j-1} \left( 1 - \alpha^{(k)} \right)$. We present the derivative by cases as follows:

$$\frac{\partial T^{(j)}}{\partial \alpha^{(i)}} = \begin{cases} 0 & (j \leq i) \\ - T^{(i-1)} \prod_{k=i+1}^{j-1} \left( 1 - \alpha^{(k)} \right) & (j > i) \end{cases}. \tag{13}$$

We combine the above expressions and yield the final result, expressed as follows:

$$\frac{\partial \hat{\boldsymbol{c}}}{\partial \alpha^{(i)}} = T^{(i)} \boldsymbol{c}^{(i)} - T^{(i-1)} \sum_{j=i+1}^{N} \left( \alpha^{(i)} \prod_{k=i+1}^{j-1} (1 - \alpha^{(k)}) \boldsymbol{c}^{(j)} \right) \tag{14}$$

$$\frac{\partial \mathcal{L}_r}{\partial \alpha^{(i)}} = T^{(i)}[\hat{c} > c_{gt}]^\top c^{(i)} - T^{(i-1)} \sum_{j=i+1}^{N} \left( \alpha^{(i)} \prod_{k=i+1}^{j-1} (1 - \alpha^{(k)})[\hat{c} > c_{gt}]^\top c^{(j)} \right) \qquad (15)$$

## B.2 Sampling Strategy

To support the discussion in the main paper regarding the magnitude variation of the network's output induced by different sampling strategies, we provide a detailed description of the sampling strategies adopted in existing methods (Wimbauer et al., 2023; Han et al., 2024; Li et al., 2024a; Feng et al., 2025) and ours. These strategies are categorized as ray sampling and point sampling.

### B.2.1 Ray Sampling

The ray sampling strategy differs between the training and evaluation phases. During training, to reduce computational cost and accelerate network convergence, we adopt a patch sampling strategy. Specifically, we extract 64 image patches (resolution: $8 \times 8$ pixels), resulting in 4,096 sampled pixels and their corresponding rays per training iteration. This sampling strategy ensures both the spatial diversity and similarity among sampled rays, thereby facilitating network training. During evaluation, we sample all pixels from the input image to infer occupancies across the entire spatial space in the camera frustum.

### B.2.2 Point Sampling

After sampling rays from the input image, points are sampled along each ray for network inference. Owing to the varying depth sensitivity of pinhole cameras, existing strategies do not sample points uniformly in the Euclidean depth space. Instead, they apply uniform sampling in the inverse-depth space, which allows for denser sampling near the camera and sparser sampling in distant areas. The distance between the camera origin and the $i$-th sampled point on a sampled ray is expressed as follows:

$$t^{(i)} = 1 / \left( \left( 1 - \frac{i+r}{N} \right) \frac{1}{t_n} + \frac{i+r}{N} \frac{1}{t_f} \right), \qquad (16)$$

where $N$ denotes the number of points sampled per ray, $r \sim N(-0.5, 0.5)$ denotes a random variable from a uniform distribution, and $t_n$ and $t_f$ represent the near and far bounds in the rendering field, respectively. The sampled points are defined as $x_c^{(i)} = o + t^{(i)} d$, where $o$ denotes the camera origin and $d$ represents the unit direction vector of the sampled ray. As discussed in the main paper, this non-uniform sampling strategy introduces variations in the network output magnitudes due to the depth-dependent sampling density of the points. During evaluation, instead of directly using voxel centers as input points to the network, we employ the same point sampling strategy as used during training, with the only difference being the removal of the random variable $r$. This eliminates the inconsistency in the spatial distribution of sampled points across both phases.

## B.3 Coordinate-Transformed Sampling Details

In the coordinate-transformed occupancy sampling process, we transform both the opacity values inferred by the network and the voxel center points from the CCS to the TCS. This transformation enables us to compute the occupancy probability for each voxel using the grid sampling algorithm. Specifically, we perform 3D grid sampling in the normalized spanning cube $[0, 1]^3$ using the bilinear interpolation mode and the border padding mode.

# C Technical Details

## C.1 Training Details

### C.1.1 Hyperparameters and Configurations

The batch size is set to 16, and the network is trained for 25 epochs. We set the loss weights, $\lambda_r$ and $\lambda_p$, to 1 and $1 \times 10^{-3}$, respectively. Furthermore, since the occlusion-aware occupancy polarization mechanism directly utilizes color similarity as a prior to infer the geometric structure between

neighboring sampled points, it is reasonable to omit color augmentation for training. Additionally, we employ horizontal flip augmentation by randomly flipping the input image prior to feeding it into the network. To preserve the original geometric structure of the scene, a corresponding inverse flip is applied to the resulting feature maps before they undergo further processing.

### C.1.2 DATASET

We train the network with both the stereo perspective-camera video sequence and the left and right fisheye-camera video sequences from the KITTI-360 dataset (Liao et al., 2022). For each fisheye image, we follow the resampling process described in Wimbauer et al. (2023) to obtain the corresponding image using a virtual perspective camera. To enlarge the overlap among camera frustums, we offset fisheye-camera sequences by ten timestamps relative to the stereo perspective-camera sequence, thereby enhancing the ratio of valid color samples.

## C.2 BENCHMARK DETAILS

### C.2.1 DATASET ALIGNMENT

The SSCBench-KITTI-360 dataset (Li et al., 2024b) provides 3D occupancy ground truth along with the corresponding 2D images. However, it employs a non-public method to select 2D image frames from the KITTI-360 dataset, resulting in updated image indices. As a result, the provided occupancy ground truth cannot be directly aligned with the test images from the KITTI-360 dataset. To address this issue, we systematically scan both datasets and construct a frame correspondence lookup table by identifying exact matches between 2D images.

### C.2.2 TRANSFORMATION BETWEEN COORDINATE SYSTEMS

In the SSCBench-KITTI-360 dataset, ground-truth 3D occupancy annotations are provided at a lower temporal resolution to reduce storage cost. Specifically, for every five consecutively reindexed frames, only one frame is associated with a 3D occupancy label. Fortunately, based on our experimental verification, all selected frames are accompanied by ground-truth poses from the original KITTI-360 dataset. This enables us to compute the relative pose transformation between a query frame in the test split and its nearest adjacent frame with ground-truth 3D occupancy annotations. Consequently, we can derive the transformation from the voxel coordinate system of the annotated occupancy grid to the camera coordinate system of the test frame.

Figure 6: An illustration of the coordinate system transformation is provided in the bird's-eye view of the occupancy ground truth. Specifically, it depicts the transformation from the voxel coordinate system associated with the $j$-th frame to the camera coordinate system of the $i$-th frame. This transformation enables subsequent computation of the frustum mask and the visibility mask.

Specifically, for the $i$-th index in the test split, we first identify the nearest subsequent index $j$ such that the corresponding frame has an associated 3D occupancy annotation in the SSCBench-KITTI-360 dataset. For example, the first test frame (index: 386) in the KITTI-360 dataset lacks an associated 3D annotation, but its nearest subsequent frame (index: 387) has a 3D occupancy label, which corresponds to index 295 in the SSCBench-KITTI-360 dataset. This mapping ensures evaluation even when direct 3D annotations are unavailable for specific test frames. Let $T_i$ and $T_j$ denote the ego poses at indices $i$ and $j$, respectively. The transformation from the LiDAR coordiante system to the camera coordinate system provided by the KITTI-360 dataset is denoted by $T_{l \to c}$, while the transformation from the voxel coordinate system to the LiDAR coordinate system provided by the SSCBench-KITTI-360 dataset is denoted by $T_{v \to l}$. $T_{v \to c}$, the transformation from the voxel coordinate system to the camera coordinate system can be obtained using the following expression:

$$T_{v \to c} = T_i^{-1} T_j T_{l \to c} T_{v \to l}. \qquad (17)$$

### C.2.3 MASK GENERATION

Based on the transformation defined in Eq. 17, we construct the frustum mask $M_f$ and the visibility mask $M_v$ corresponding to the 3D occupancy ground truth. For each voxel center $v_c^{(i)}$, we define binary values $m_f^{(i)}$ and $m_v^{(i)}$ in this voxel to represent its frustum and visibility status, respectively. Given the projected image coordinates $(u, v)$ that correspond to the voxel center $v_c^{(i)}$, the value in the frustum mask is defined as follows:

$$m_f^{(i)} = [0 \le u \le w - 1] \wedge [0 \le v \le h - 1], \tag{18}$$

where $w$ and $h$ denote the image width and height, respectively. In this equation, $m_f^{(i)} = 1$ indicates that the voxel projects within the valid image bounds.

We apply a ray tracing algorithm to generate visibility masks based on 3D occupancy ground truth. Specifically, we generate a ray for each image pixel and perform dense sampling along the ray within the spatial bounds of the 3D occupancy volume. The sampling interval is set equal to the voxel size to ensure that any two adjacent sampled points lie either within the same voxel or in adjacent voxels. Subsequently, we query the ground-truth occupancy at each sampled location to determine whether it is occupied. For a given ray, the visibility status $m_v^{(i)}$ of a point is set to 1 if the point itself and all preceding points along the ray are unoccupied. These per-point visibility statuses are then mapped back to their corresponding voxels, resulting in the final voxel-wise visibility mask $M_v$.

### C.2.4 OCCUPANCY METRICS DETAILS

The expressions of the evaluation metrics used in the main paper is given as follows:

$$O_{\text{Acc}} = \frac{\sum_i \left[\hat{o}^{(i)} = o^{(i)}\right] m_f^{(i)}}{\sum_i m_f^{(i)}}, \tag{19}$$

$$O_{\text{Pre}} = \frac{\sum_i o^{(i)} \hat{o}^{(i)} m_f^{(i)}}{\sum_i \hat{o}^{(i)} m_f^{(i)}}, \tag{20}$$

$$O_{\text{Rec}} = \frac{\sum_i \hat{o}^{(i)} o^{(i)} m_f^{(i)}}{\sum_i o^{(i)} m_f^{(i)}}, \tag{21}$$

$$\text{IE}_{\text{Acc}} = \frac{\sum_i \left[\hat{o}^{(i)} = o^{(i)}\right] (1 - m_v^{(i)}) m_f^{(i)}}{\sum_i (1 - m_v^{(i)}) m_f^{(i)}}, \tag{22}$$

$$\text{IE}_{\text{Pre}} = \frac{\sum_i (1 - o^{(i)})(1 - \hat{o}^{(i)})(1 - m_v^{(i)}) m_f^{(i)}}{\sum_i (1 - \hat{o}^{(i)})(1 - m_v^{(i)}) m_f^{(i)}}, \tag{23}$$

$$\text{IE}_{\text{Rec}} = \frac{\sum_i (1 - \hat{o}^{(i)})(1 - o^{(i)})(1 - m_v^{(i)}) m_f^{(i)}}{\sum_i (1 - o^{(i)})(1 - m_v^{(i)}) m_f^{(i)}}, \tag{24}$$

where $\hat{o}^{(i)}$ and $o^{(i)}$ denote the predicted and ground-truth occupancies for the $i$-th voxel, respectively.

## D ADDITIONAL EXPERIMENTAL RESULTS

### D.1 OCCLUSION-AWARE OCCUPANCY POLARIZATION

When designing the polarization mechanism, our goal is to establish an indicator that reflects whether two adjacent 3D points along a ray correspond to the same object when projected into a

| Baseline | Signal for building $\mathcal{L}_p$ | $O_{Acc}$ | $IE_{Acc}$ | $IE_{Rec}$ |
|---|---|---|---|---|
| BTS | Pseudo depth | 0.879 | 0.734 | 0.644 |
| | RGB intensity | **0.880** | **0.737** | **0.667** |
| KDBTS | Pseudo depth | 0.878 | 0.724 | 0.679 |
| | RGB intensity | **0.879** | **0.725** | **0.682** |
| ViPOcc | Pseudo depth | 0.881 | 0.740 | 0.655 |
| | RGB intensity | **0.883** | **0.741** | **0.676** |

Table 6: Ablation study on the signals of occlusion-aware occupancy polarization mechanism.

given source view. Other than relying on the color difference, we exploit the difference in pseudo depth predicted by a vision foundation model (Yang et al., 2024). Specifically, for each pair of adjacent samples along a ray, we obtain their projections in the source view and examine the discrepancy of their image-level signals (*e.g.*, color or pseudo depth) as an indicator to determine whether these projections lie on the same object.

Experimental results presented in Table 6 suggest that RGB-based indicators consistently outperform pseudo-depth-based ones across baseline models. This finding is somewhat unexpected, given that depth maps typically provide more reliable geometric cues and are generally more robust to texture ambiguity and color similarity.

We hypothesize that this phenomenon stems from the distinct signal transition characteristics across object boundaries. For two adjacent points located on different objects, the pseudo-depth differences can vary significantly depending on the geometric structure of the scene. In particular, when these points lie on different objects but have similar depth values, their pseudo-depth values are often close, making it difficult to distinguish inter-object cases from intra-object ones. In contrast, due to variations in lighting, material, and texture across different objects, RGB intensities tend to differ markedly across object boundaries, even when depth values are similar. This observation suggests that RGB intensity is generally more reliable than pseudo-depth for inferring object-level consistency. In the future, we plan to further optimize the design of the polarization mechanism by developing a lightweight object-consistency indicator that does not rely on vision foundation models or additional ground truth annotations.

### D.2 3D Occupancy Prediction Visualization

We present additional qualitative results of 3D occupancy prediction on the KITTI-360 dataset. As shown in Fig. 7, our method achieves superior geometric reconstruction performance in occluded regions compared to previous state-of-the-art approaches. This improvement can be attributed to the proposed occlusion-aware occupancy polarization mechanism, which effectively leverages complementary visual cues from alternative viewpoints to recover missing structural information.

## E LIMITATIONS

Despite achieving SoTA performance, the proposed method still presents a known limitation related to the camera field of view (FOV). During training, forward-view cameras at frame $t$ and side-view cameras at frame $t + T$ are selected (where $T$ is a fixed parameter), and the supervisory signals are primarily derived in regions where their FOVs overlap. As illustrated in Fig. 8, the narrow horizontal FOV of the cameras results in a limited overlapping volume, thereby weakening the supervision available during training. Our future work will investigate improved dataset organization strategies to increase the volume of overlapping regions.

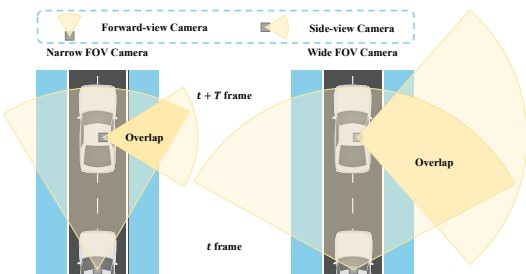

Figure 8: An illustration of the limitation related to camera field of view.

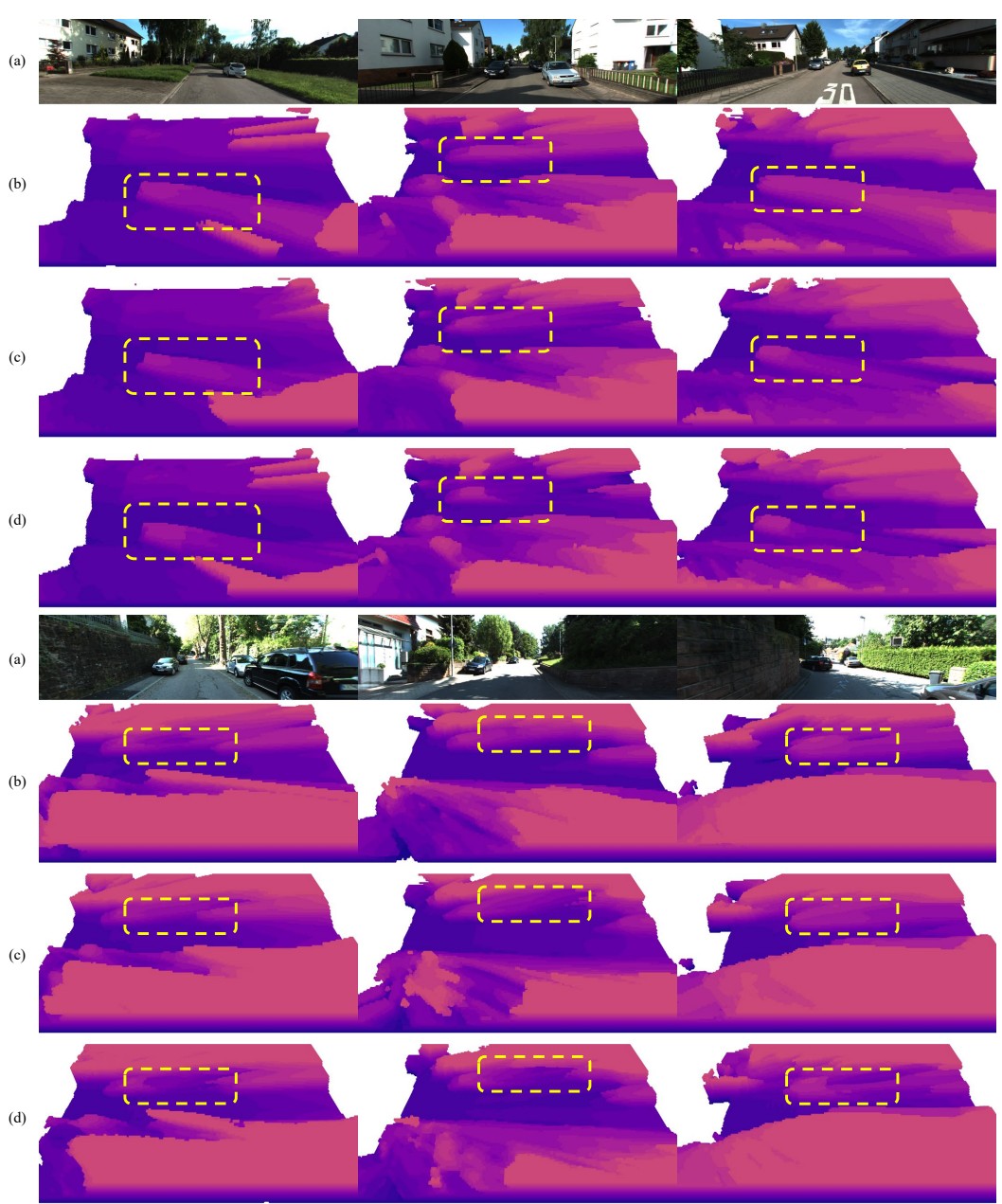

Figure 7: Qualitative comparison of occupancy prediction on KITTI-360 dataset: (a) RGB images; (b) BTS (Wimbauer et al., 2023) results; (c) ViPOcc (Feng et al., 2025) results; (d) our results.

