# OpenReview forum: "Rebenchmarking Unsupervised Monocular 3D Occupancy Prediction"
_ICLR.cc/2026/Conference — Submitted to ICLR 2026_

### Official Review · Reviewer_mswk · 2025-10-26

**Soundness:** 2
**Presentation:** 3
**Contribution:** 2
**Rating:** 4
**Confidence:** 5

**Summary:**

This paper proposes an interpretable, opacity-based occupancy representation and introduces a coordinate-transformed sampling algorithm to align predictions with 3D voxel annotations.
Furthermore, it designs an occlusion-aware occupancy polarization mechanism that leverages multi-view color discrepancies to provide supervisory signals for occluded regions.
Extensive experiments on KITTI-360 validate the proposed benchmark's rationality and demonstrate good performance, showing competitive results against supervised methods.

**Strengths:**

1. The paper is well-written and easy to follow. The visual demonstration is also good.
2. The performance of the method is good compared with the single-view based methods.
3. The proposed occlusion-aware occupancy polarization is novel which effectively imposes constraints to occluded regions with different colors along the sampling ray.

**Weaknesses:**

1. Flawed analysis in the method section. Please see the first three questions in the Questions section.
2. Limited comparison. The paper does not compare with multi-view unsupervised occupancy prediction methods which can readily be applied for single-view task, such as SelfOcc.

**Questions:**

1. In Line 194-195, the paper claims that the rendering contributions of points B and C are theoretically the same. Further explanation is needed here since in rendering process, C is occluded by B and thus contributes less to the final result.
2. In Line 169-170, the paper claims that existing methods  adopt a fixed threshold of 0.5 to binarize each voxel. I also think this is unreasonable, but the paper should provide proper reference work here.
3. In Line 198-200, the paper analyzes the correlation between sampling interval and density value based on the assumption that the rendering contributions of points B in densely sampled region and C in sparsely sampled region should be the same. However, I think the assumption is incorrect, since densely sampled region would have more individual samples and thus the contribution of a single point in such a region does not have to comparable to that of a point in the sparsely sampled region. After all, the rendering process is a integration along the line.
4. What is the advantage of the proposed method compared with SDF-based representation without the need for a handcrafted threshold, and with methods based on 3D representations such as BEV / TPV without the need for post grid sampling?

---

> ### Author Response · Authors · 2025-11-26
>
> We thank Reviewer mswk for their thorough and insightful feedback on methodology analysis and limited comparisons, and we address all these concerns below.
>
> W1 & Q1 & Q3: Method Analysis Questions
>
> Q1: First, we would like to clarify the notation in Figure 1: **the solid segments denote the occupied portions of the sampled rays, while the dashed segments indicate the portions inside an object** (we add a legend for this in the revised figure). Accordingly, for the blue ray, points B and C both lie *below* the road surface, instead of on the object boundaries. Thus, both B and C have nearly identical accumulated transmittance values $T$ that decay to approximately 0. Therefore, equal opacities $\alpha_B = \alpha_C = 1$ leads to approximately equal rendering contributions $\alpha T$.
>
> Based on the above analysis, the only factor that differs between B and C is the sampling interval $\delta$: B lies in a more densely sampled region with $\delta_B < \delta_C$. Using Eq.2, the relationship $\alpha_B = \alpha_C = 1$ directly yields $\sigma_B > \sigma_C$. This analysis, therefore, characterizes the inequality of $\sigma$ induced by the discretized rendering formulation.
>
> Q3: Regarding the rendering weights, the reviewer’s concern appears to stem from interpreting the rendering contribution $\alpha T$ of an interval as that of a single point.
> However, as shown in Eq.2, the opacity $\alpha$ explicitly derives from the interval size $\delta$. Consequently, both $\alpha$ and rendering weight represent the property of a finite spatial neighborhood, ensuring physical consistency regardless of sampling density. Therefore, the rendering weight of a spatial neighborhood is not determined by the sampling density.
>
> Q2: Reference
>
> The first existing method adopting a fixed threshold of 0.5 is BTS[1]. Relevant details are provided in Supplement Sec 4.3. Subsequent methods[2][3][4] follows this approach.
>
> - [1]: Wimbauer, et al. "Behind the Scenes: Density Fields for Single View Reconstruction" CVPR'23
> - [2]: Han, et al. "Boosting Self-Supervision for Single-View Scene Completion via Knowledge Distillation" CVPR'24
> - [3]: Li, et al. "Know Your Neighbors: Improving Single-View Reconstruction via Spatial Vision-Language Reasoning" CVPR'24
> - [4]: Feng, et al. "ViPOcc: Leveraging Visual Priors from Vision Foundation Models for Single-View 3D Occupancy Prediction" AAAI'25
>
> W2: We provide quantitative results on the SemanticKITTI dataset against SelfOcc as follows. Our method outperforms both forms for SelfOcc in IoU metrics. For more details, please refer to Sec. 4.2 paragraph 3 in the revised paper.
>
> | Method | IoU (%) | Pre (%) | Rec (%) |
> | --- | --- | --- | --- |
> | SceneRF | 13.8 | 17.3 | 41.0 |
> | SelfOcc(BEV) | 21.0 | 37.3 | 32.4 |
> | SelfOcc(TPV) | 22.0 | 34.8 | 37.3 |
> | ViPOcc | 23.6 | 26.9 | 66.8 |
> | Ours | 24.1 | 27.0 | 68.7 |

---

> > ### Author Response · Authors · 2025-11-26
> >
> > Q4: Comparison with Other Methods
> >
> > We would first like to clarify that 0.5 is not a *handcrafted* threshold for opacity $\alpha$. For the SDF-based representation, the output $s$ is defined over the range $(-\infty, +\infty)$, and 0 is used as the threshold to determine occupancy. Similarly, for our proposed occupancy probability representation, whose values lie within $(0,1)$, using the midpoint 0.5 as the threshold for occupancy binarization is a natural and principled choice. These thresholds are not manually set, but are fixed values determined by the respective ranges of both occupancy representations.
> >
> > Having clarified the threshold issue, we now discuss the advantages of NeRF-based methods over SDF-based approaches. SelfOcc first converts the SDF to opacity $\alpha$ and then applies the NeRF volume rendering formula to produce the rendered images.
> > In terms of the main supervisory signal, SDF-based methods like SelfOcc still rely on volume rendering, as do NeRF-based methods.
> >
> > However, NeRF-based methods are simpler in the design of both model architecture and auxiliary loss functions compared to SDF-based methods like SelfOcc:
> >
> > - They do not require additional network modules (like the 3D encoder in SelfOcc) and are therefore more lightweight than SDF-based approaches;
> > - They do not require extra auxiliary loss functions, which are necessary in SDF-based methods. For instance, SelfOcc (Sec. 3.3) introduces many SDF regularization constraints, whereas NeRF-based methods avoid them, reducing the complexity of hyperparameter tuning.
> >
> > We would like to clarify that the proposed post grid sampling step during the inference phase is not redundant for NeRF-based approaches. Instead, it is a necessary procedure for converting the radially distributed occupancy predictions into a voxel-wise representation. Without this procedure, TPV/BEV methods sacrifice the spatial flexibility of 3D occupancy prediction, as they rely on handcrafted configurations such as predefined voxel resolutions and spatial perception ranges.
> >
> > Furthermore, unlike BEV/TPV approaches, NeRF-based methods do not require operations on 3D features or feature-dimension lifting, which significantly reduces both the number of network parameters and the overall training complexity.

---

### Official Review · Reviewer_TgT8 · 2025-10-31

**Soundness:** 3
**Presentation:** 2
**Contribution:** 3
**Rating:** 6
**Confidence:** 2

**Summary:**

This paper revisits and reformulates the evaluation protocols for unsupervised monocular 3D occupancy prediction—an increasingly important task in vision-centric autonomous driving. The authors systematically dissect the relationship between NeRF-based network outputs and voxel-wise occupancy ground truth, arguing that opacity ($\alpha$), rather than density ($\sigma$), yields a more robust and physically meaningful basis for evaluation. They propose a coordinate-transformed sampling algorithm to align opacity predictions with voxel grids, enabling fair benchmarking against supervised methods. Additionally, they introduce an occlusion-aware occupancy polarization loss, leveraging multiview cues to sharpen occupancy predictions in occluded regions. Experimental results on the KITTI-360 dataset show improved alignment between training and evaluation, competitive or superior performance over existing unsupervised and even some supervised baselines, and substantial advances in occlusion reasoning.

**Strengths:**

1. The paper thoughtfully revisits evaluation protocols and provides a clear, equation-rich justification for why opacity ($\alpha$) is preferable to network density ($\sigma$) for occupancy probability.
2. The methodology is generally well explained, algorithms are clearly laid out, supplemental details/figures are available, and the quantification of evaluation regions enhances reproducibility and interpretability.
3. Robust Experimental Suite: Quantitative and ablation results on KITTI-360 provide strong support for the method’s claims.

**Weaknesses:**

1. All results are on a single dataset (KITTI-360), and there is no exploration of other driving, indoor, or multi-modal datasets.  This raises questions about generalization and the broader applicability of the proposed evaluation protocol and methods.  Given the benchmark’s ambition for field-wide adoption, a demonstration on at least one additional, differently-distributed dataset would have been highly appropriate.

**Questions:**

1. How would the coordinate-transformed sampling and polarization loss perform on non-driving (e.g., indoor, or synthetic) datasets?  Are there failure cases or calibration routines required for generalization?
2. How sensitive are the results (especially occlusion region accuracy) to the choices of $\lambda_r, \lambda_p$, and threshold values for opacity?  Are there guidelines for robust tuning across differing scenes?

---

> ### Author Response · Authors · 2025-11-26
>
> We thank Reviewer TgT8 for their insightful review about scene generalization and hyperparameter tuning, and we address all these concerns below.
>
> W1 & Q1: Scene Generalization
>
> As stated in the first sentence of our abstract, our work focuses on ***vision-centric autonomous driving***, aiming to address the fundamental challenge of 3D structure inference in this domain. The application of our method to non-driving scenarios is outside the scope of this work. Nevertheless, we additionally provide an analysis of the method’s applicability to broader scenarios and further present experiments in other driving settings.
>
> The proposed coordinate-transformed sampling algorithm is designed to address the spatial misalignment problem between the network output and the voxel-wise ground truth, and its formulation is inherently scenario-independent.
> In addition, the polarization loss is derived from the multi-view consistency assumption and can be generalized to arbitrary 3D spaces.
> Consequently, both components can apply to indoor and outdoor environments, as well as real and synthetic datasets.
>
>
> To further demonstrate the generalizability in driving scenarios, we provide quantitative results on the SemanticKITTI dataset (with a similar FOV to that of the KITTI-360 dataset) as follows. Our method outperforms previous SoTA methods, achieving 2% and 3% improvements in IoU and recall, respectively.
> Following the reviewer's suggestions, we have presented this experimental results in the revised paper. Please refer to Sec. 4.2 paragraph 3.
>
> | Method | IoU (%) | Pre (%) | Rec (%) |
> | --- | --- | --- | --- |
> | SceneRF | 13.8 | 17.3 | 41.0 |
> | SelfOcc(BEV) | 21.0 | 37.3 | 32.4 |
> | SelfOcc(TPV) | 22.0 | 34.8 | 37.3 |
> | ViPOcc | 23.6 | 26.9 | 66.8 |
> | Ours | 24.1 | 27.0 | 68.7 |
>
> We also conduct experiments on nuScenes and Waymo datasets. To be specific, in each training iteration we use the front-camera image at frame $t$ together with the back-left, back-right camera images at frame $t+6$. The offset of 6 frames is an empirical choice to maximize the spatial overlap between views. However, current unsupervised 3D occupancy prediction methods (BTS, KDBTS, KYN, ViPOcc & Ours) struggle to converge. We attribute this phenomenon to the narrow camera field of view in these datasets.
>
> Current unsupervised 3D occupancy prediction methods fundamentally rely on multi-view consistency constraints. As discussed in Sec. 3.4, all supervisory signals are derived from multi-view 2D images and supervise the network’s occupancy predictions only within regions where camera views overlap. Consequently, the volume of these overlapping regions, which is positively correlated with the cameras' horizontal field of view (FOV), directly determines the effectiveness of the training process.
> However, compared with the KITTI-360 dataset (FOV = 103°), nuScenes (FOV = 64°) and Waymo (FOV = 50°) provide much narrower camera views. The resulting low proportion of overlapping regions severely hinders the establishment of multi-view supervisory signals, preventing the metrics from converging.
> To further validate this phenomenon, we reduce the FOV of KITTI-360 images to 64°, matching that of nuScenes, and observe the same issue: the network’s performance metrics cease to converge. Additional details and illustrative figures are provided in Sec. E in the revised paper.
>
> Q2: Hyperparameter Sensitivity
>
> Since $\lambda_r$ serves as the primary loss weight and is conventionally fixed at 1 without affecting training stability, we vary only $\lambda_p$ to analyze its influence on occlusion-region accuracy:
>
> | $\lambda_p$ | $\mathrm{O}_{\text{Acc}}$ | $\mathrm{IE}_{\text{Acc}}$ | $\mathrm{IE}_{\text{Rec}}$ |
> | -- | -- | -- | -- |
> | $1 \times 10^{-4}$ | 0.880 | 0.740 | 0.654 |
> | $5 \times 10^{-4}$ | 0.879 | 0.737 | 0.661 |
> | $1 \times 10^{-3}$ | 0.883 | 0.741 | 0.676 |
> | $5 \times 10^{-3}$ | 0.880 | 0.737 | 0.654 |
> | $1 \times 10^{-2}$ | 0.878 | 0.734 | 0.651 |
>
> We range $\lambda_p$ from $10^{-4}$ to $10^{-2}$.
> The algorithm achieves best performance when $\lambda_p=1\times 10^{-3}$ and the changes in $O_{Acc}$, $IE_{Acc}$, and $IE_{Rec}$ with respect to $\lambda_p$ do not exceed 0.5%, 0.9%, and 3.6% respectively.
>
> We want to clarify that 0.5 is not a *handcrafted* threshold for our proposed occupancy probability representation. The probability is defined over $(0,1)$, and 0.5 is a fixed value determined by the range of the representation.
>
> When we release the code, we will provide users with a detailed guide to code reproduction and model fine-tuning.

---

### Official Review · Reviewer_zWZc · 2025-10-31

**Soundness:** 3
**Presentation:** 2
**Contribution:** 3
**Rating:** 6
**Confidence:** 4

**Summary:**

This paper addresses critical limitations in existing unsupervised monocular 3D occupancy prediction, primarily the inconsistency between training and evaluation protocols and poor performance in occluded regions. It proposes an opacity-based occupancy probability representation to replace the scale-sensitive network output, resolving magnitude variation issues. A coordinate-transformed occupancy sampling algorithm is proposed to align radial opacity distributions with voxel-wise 3D ground truth. Furthermore, it proposes an occlusion-aware occupancy polarization mechanism using multi-view visual cues to enhance supervision in occluded areas. Experiments on KITTI-360 show the method outperforms unsupervised SOTA and matches supervised methods, while establishing a unified 3D benchmark.

**Strengths:**

1. This method identifies the inconsistency between point-wise rendering weight outputs and voxel-wise ground truth in existing NeRF-based methods, and uses opacity to resolve this, improving evaluation reliability.
2. The coordinate-transformed sampling effectively bridges the spatial gap between radial opacity and uniform voxel grids, enabling direct comparison between unsupervised and supervised methods.
3. The occlusion-aware polarization mechanism leverages color differences between adjacent points to supplement supervision in occluded regions.

**Weaknesses:**

1. The benchmark relies solely on the KITTI-360 dataset. No experiments are conducted on other datasets, such as nuScenes, to verify the method’s generalizability to different driving scenarios.
2. Qualitative results for occluded regions lack dedicated quantitative metrics, making it hard to objectively assess the polarization mechanism’s improvement on occlusion reasoning. Additionally, there is no clear explanation as to whether the improved ability of occlusion reasoning contributes to the safety of real-world autonomous driving.
3. Since the paper criticizes KYN for its high computational cost due to visual-language networks, it should provide a quantitative analysis of its own method’s inference efficiency.

**Questions:**

See weakness

---

> ### Author Response · Authors · 2025-11-26
>
> We thank Reviewer zWZc for their critical concerns regarding generalizability, occlusion problems, and inference efficiency, and we address all of these concerns below.
>
> W1: Limited Dataset
>
> Following the reviewer's suggestion, we conduct experiments on nuScenes and Waymo datasets. To be specific, in each training iteration we use the front-camera image at frame $t$ together with the back-left, back-right camera images at frame $t+6$. The offset of 6 frames is an empirical choice to maximize the spatial overlap between views. However, current unsupervised 3D occupancy prediction methods (BTS, KDBTS, KYN, ViPOcc & Ours) struggle to converge. We attribute this phenomenon to the narrow camera field of view in these datasets.
>
> Current unsupervised 3D occupancy prediction methods fundamentally rely on multi-view consistency constraints. As discussed in Sec. 3.4, all supervisory signals are derived from multi-view 2D images and supervise the network’s occupancy predictions only within regions where camera views overlap. Consequently, the volume of these overlapping regions, which is positively correlated with the cameras' horizontal field of view (FOV), directly determines the effectiveness of the training process.
> However, compared with the KITTI-360 dataset (FOV = 103°), nuScenes (FOV = 64°) and Waymo (FOV = 50°) provide much narrower camera views. The resulting low proportion of overlapping regions severely hinders the establishment of multi-view supervisory signals, preventing the metrics from converging.
> To further validate this phenomenon, we reduce the FOV of KITTI-360 images to 64°, matching that of nuScenes, and observe the same issue: the network’s performance metrics cease to converge. Additional details and illustrative figures are provided in Sec. E in the revised paper.
>
> To further demonstrate the generalizability, we provide quantitative results on the SemanticKITTI dataset (with a similar FOV to that of the KITTI-360 dataset) as follows. Our method outperforms previous SoTA methods, achieving 2% and 3% improvements in IoU and recall, respectively.
> Following the reviewer's suggestions, we have presented this experimental results in the revised paper. Please refer to Sec. 4.2 paragraph 3.
>
> | Method | IoU (%) | Pre (%) | Rec (%) |
> | --- | --- | --- | --- |
> | SceneRF | 13.8 | 17.3 | 41.0 |
> | SelfOcc(BEV) | 21.0 | 37.3 | 32.4 |
> | SelfOcc(TPV) | 22.0 | 34.8 | 37.3 |
> | ViPOcc | 23.6 | 26.9 | 66.8 |
> | Ours | 24.1 | 27.0 | 68.7 |
>
> W2: Problems for Occluded Regions
>
> We present an ablation study in Table 5 that quantitatively demonstrates the effectiveness of the proposed occupancy polarization mechanism.
> In the table, our manuscript already provides dedicated quantitative metrics for occluded regions. Specifically, $IE_{Acc}$ and $IE_{Rec}$ are metrics for the intersection between the camera frustum and the invisibility mask (occluded regions), as described in lines 365-366. More details are provided in Supplement C.2.3 & C.2.4.
> As shown in Table 5, the incorporation of the polarization mechanism yields maximum improvements of 1.8% and 1.4% on $IE_{Acc}$ and $IE_{Rec}$, respectively. These results are detailed in the revised paper.
>
> While we do not directly evaluate real-world autonomous driving safety, our work addresses the fundamental scientific problem of unsupervised 3D occupancy prediction under occlusion. As shown in Figure 4, our method effectively mitigates trailing effects, indicating superior ability to recover the shapes of 3D object instances, thereby strengthening the system's recognition and perception of traffic participants in the environment. This improvement in accurate 3D perception can ultimately benefit safer decision-making in autonomous driving systems.
>
> W3: Inference Efficiency Analysis
>
> Following the reviewer's suggestion, we provide the quantitative analysis of the current unsupervised methods' inference efficiency as follows:
>
> | Method | Parameters (M) | Inference Speed (FPS) |
> | --- | --- | --- |
> | KYN | 530.5 | 0.54 |
> | Ours | 43.9 | 2.38 |
>
> As shown in the table, KYN has more network parameters and a slower inference speed than ours.

---

### Official Review · Reviewer_xg3Y · 2025-11-01

**Soundness:** 2
**Presentation:** 2
**Contribution:** 2
**Rating:** 4
**Confidence:** 3

**Summary:**

Addressing the evaluation inconsistencies and occlusion modeling challenges in unsupervised monocular 3D occupancy prediction, this paper presents three core contributions: 1) Reinterpreting occupancy probability in neural radiance fields (NeRF) by replacing density σ with opacity α to resolve sampling-dependency issues; 2) Designing a coordinate-transformed occupancy sampling algorithm to align the distribution of α with voxel-wise ground truth, unifying the evaluation space for both unsupervised and supervised methods; 3) Proposing an occlusion-aware occupancy polarization mechanism that leverages multi-view color cues to supplement supervision in occluded regions. On the KITTI-360 dataset, the proposed method outperforms SOTA unsupervised methods and achieves intersection over IoU values comparable to or even exceeding some supervised methods. Ablation studies validate the rationality of each module. The source code and evaluation protocol will be made publicly available upon publication.

**Strengths:**

1. Clarity of Presentation: The paper is well-structured with clear explanations of concepts, methodologies, and experimental results.
2. Significance of Benchmark Construction: Constructing a benchmark for monocular 3D occupancy prediction addresses a critical gap in the field. As unsupervised monocular 3D occupancy prediction is essential for vision-centric autonomous driving, a dedicated benchmark contributes to standardized evaluation and fair comparison of subsequent methods, which is of great practical value.

**Weaknesses:**

1. **Limited Dataset Evaluation:** The paper only conducts experiments on the KITTI-360 dataset and lacks evaluation on mainstream autonomous driving datasets such as nuScenes. Mainstream datasets like nuScenes cover more complex scenarios, and evaluating only on KITTI-360 fails to demonstrate the generalizability of the proposed benchmark and method. This limits the reliability of the work’s conclusions regarding real-world applicability.
2. **Insignificant Performance Advantages:** As shown in Table 2, the proposed method does not significantly outperform other existing methods (e.g., ViPOcc). This weakens the persuasiveness of the method’s effectiveness, as readers cannot clearly perceive the added value of the proposed innovations compared to prior works.
3. **Lack of Coherent Contributions:** The research problems addressed in the paper lack a coherent main thread. The three proposed contributions are presented as relatively independent modules, failing to clearly articulate the core overarching issue the paper intends to solve. The work is overly engineering-oriented, with insufficient emphasis on the underlying scientific questions and systematic design logic of the benchmark.

**Questions:**

All questions are detailed in the "Weaknesses" section above.

---

> ### Author Response · Authors · 2025-11-26
>
> We thank Reviewer xg3Y for the constructive feedback on our evaluation scale, performance gains, and overall contribution coherence, and we address all these concerns below.
>
> W1: Limited Dataset Evaluation
>
> Following the reviewer's suggestion, we conduct experiments on nuScenes and Waymo datasets. To be specific, in each training iteration we use the front-camera image at frame $t$ together with the back-left, back-right camera images at frame $t+6$. The offset of 6 frames is an empirical choice to maximize the spatial overlap between views. However, current unsupervised 3D occupancy prediction methods (BTS, KDBTS, KYN, ViPOcc & Ours) struggle to converge. We attribute this phenomenon to the narrow camera field of view in these datasets.
>
> Current unsupervised 3D occupancy prediction methods fundamentally rely on multi-view consistency constraints. As discussed in Sec. 3.4, all supervisory signals are derived from multi-view 2D images and supervise the network’s occupancy predictions only within regions where camera views overlap. Consequently, the volume of these overlapping regions, which is positively correlated with the cameras' horizontal field of view (FOV), directly determines the effectiveness of the training process.
> However, compared with the KITTI-360 dataset (FOV = 103°), nuScenes (FOV = 64°) and Waymo (FOV = 50°) provide much narrower camera views. The resulting low proportion of overlapping regions severely hinders the establishment of multi-view supervisory signals, preventing the metrics from converging.
> To further validate this phenomenon, we reduce the FOV of KITTI-360 images to 64°, matching that of nuScenes, and observe the same issue: the network’s performance metrics cease to converge. Additional details and illustrative figures are provided in Sec. E in the revised paper.
>
> To further demonstrate the generalizability, we provide quantitative results on the SemanticKITTI dataset (with a similar FOV to that of the KITTI-360 dataset) as follows. Our method outperforms previous SoTA methods, achieving 2% and 3% improvements in IoU and recall, respectively.
> Following the reviewer's suggestions, we have presented this experimental results in the revised paper. Please refer to Sec. 4.2 paragraph 3.
>
> | Method | IoU (%) | Pre (%) | Rec (%) |
> | --- | --- | --- | --- |
> | SceneRF | 13.8 | 17.3 | 41.0 |
> | SelfOcc(BEV) | 21.0 | 37.3 | 32.4 |
> | SelfOcc(TPV) | 22.0 | 34.8 | 37.3 |
> | ViPOcc | 23.6 | 26.9 | 66.8 |
> | Ours | 24.1 | 27.0 | 68.7 |
>
> W2: Insignificant Performance Advantages
>
> Our work primarily focuses on a classification task that determines whether voxels in space are occupied. For such a problem, ***considering precision or recall in isolation*** may not yield a comprehensive evaluation, as discussed in Sec. 4.2, paragraph 2.
>
> By combining both precision and recall, the F1-score provides a balanced view for optimal model evaluation. F1-score and IoU are positively correlated:
> $$
> F_1 = \frac{2}{\mathrm{Pre}^{-1} + \mathrm{Rec}^{-1}} = \frac{2\mathrm{TP}}{2\mathrm{TP} + \mathrm{FP} + \mathrm{FN}} = \frac{2}{1 + \mathrm{IoU}^{-1}}
> $$
> Therefore, in Table 2, the IoU metric serves as the most important indicator. Our method improves IoU by 2% over the existing state-of-the-art (KDBTS), whereas KDBTS improves by only 0.5% over the earlier state-of-the-art (KYN). In light of this, the performance improvement of the proposed method is far more significant than that achieved in some previous works.
>
> W3: Lack of Coherent Contributions
>
> We discuss the relationship among the three contributions as follows:
>
> We first propose an interpretable representation, opacity $\alpha$, for the occupancy probability to eliminate inconsistencies between the training and evaluation protocols. However, as shown in Figure 2, the predicted opacities spatially misalign with occupancy annotations. To address this issue, we introduce our second contribution, a coordinate-transformed occupancy sampling algorithm. These two contributions establish a benchmark for reliable and interpretable evaluations of 3D occupancy prediction approaches. Further experiments on this benchmark reveal the performance degradation in occluded regions in existing methods (please refer to Table 3 and Sec 4.3, paragraph 2). To address this problem, we conduct a quantitative analysis and develop the proposed occlusion-aware occupancy polarization mechanism to effectively mitigate this limitation. Collectively, these contributions constitute a coherent pipeline that systematically advances effective, interpretable unsupervised 3D occupancy prediction in both training and evaluation.

---

### Author Response · Authors · 2025-11-26

We want to express our gratitude to the reviewers for acknowledging the strengths of our work, including:

- Clear writing (Reviewer xg3Y, Reviewer TgT8, Reviewer mswk)
- Benchmark significance (Reviewer xg3Y, Reviewer zWZc, Reviewer TgT8)
- Methodology novelty (Reviewer zWZc, Reviewer mswk)
- Superior performance (Reviewer TgT8, Reviewer mswk)

In response to the reviewers' questions, we have provided corresponding explanations and conducted further experiments under each review. We are looking forward to further discussions!

---

### Comment · Area_Chair_9g9W · 2025-11-27

Dear reviewers,

Please review the rebuttal and discuss with the authors if you have not done it.

Thanks,
AC

---

### Author Response · Authors · 2025-11-29
**Rebuttal Summary For Area Chair's Convenience**

Below, we concisely summarize the main points of our rebuttal for the Area Chair’s convenience.

1. **Limited Dataset Evaluation (Reviewers xg3Y, zWZc, TgT8).**  In response to concerns regarding dataset diversity, we conduct additional experiments on three further datasets. Results on SemanticKITTI are included in both the revised manuscript and the rebuttal. Moreover, we thoroughly analyze failure cases observed in nuScenes and Waymo, and substantiate our hypothesis regarding the influence of narrow fields of view through targeted experiments.

2. **Methodological Clarifications (Reviewers mswk, xg3Y).**  For Reviewer mswk, we augment Figure 1 with additional notations to resolve the ambiguity concerning point locations. We also reinterpret the concept of *rendering weight*, emphasizing that it is defined over a finite spatial neighborhood rather than individual sampled points, which is an issue that lies at the core of the reviewer’s misunderstanding. For Reviewer xg3Y, we clarify the interdependence among the three proposed contributions and elaborate on their underlying causality and intuition.

3. **Concerns on Performance Advantages (Reviewers mswk, xg3Y).**  For Reviewer xg3Y, we provide quantitative evidence demonstrating improvements in key metrics and highlight that the gains achieved by our method substantially exceed those reported in prior work. For Reviewer mswk, we expand the comparison against SelfOcc and other unsupervised multi-view approaches, and further analyze the strengths of our method relative to SDF-based methods and BEV/TPV representations.

4. **Experimental Details (Reviewers zWZc, TgT8).**  For Reviewer zWZc, we explicitly point out the dedicated quantitative metrics used for occluded regions to address the reviewer’s confusion. We also provide computational cost statistics, as the reviewer suggested. For Reviewer TgT8, we include a sensitivity analysis on key hyperparameters based on newly conducted experiments.

We believe that these revisions and clarifications substantively address the reviewers’ concerns, and we sincerely appreciate the Area Chair’s time and consideration.

---

### Meta-Review · Area_Chair_Wp4N · 2026-01-06

**Summary:**

The reviewers acknowledge the paper’s timely focus on the occluded geometry problem in self-supervised monocular 3D occupancy prediction, which is a critical but often overlooked limitation of photometric-consistency-based learning. However, several core concerns were raised during the discussion, including the generality of the findings across datasets (like Waymo and nuScenes); and the performance improvement over the SOTA approaches. The authors supplemented with additional results from other approaches like SelfOcc, and demonstrated the superiority of the derived approach.

**Reviewer Concerns:**

Although there has been no further discussion from the reviewers after the authors' response, I personally think the clarifications on technical details effectively address the reviewers' questions. However, as more experimental results have been provided, it becomes clearer that the new metrics remain positively correlated with IoU; if the rankings do not change and the conclusions are not reversed, the "new" metrics may not offer unique value. Additionally, the metrics themselves may not show their power in narrow FoV settings.

**Reviewer Scores:**

There is no discussion from reviewers after the authors' response. I personally think the clarifications on technical details can well address the reviewers' questions. However, for the reviewers who raised concerns about dataset diversity, they might not be fully satisfied. An exception is Reviewer TgT8, whose questions were very specific and the authors provided very specific answers; therefore, TgT8 is likely to increase the score.

---

### Decision · Program_Chairs · 2026-01-26

Reject